Host-Microbe Biology
# Microbial Strategies for Survival in the Glass Sponge *Vazella pourtalesii*

Kristina Bayer,[a] Kathrin Busch,[a] Ellen Kenchington,[b] Lindsay Beazley,[b] Sören Franzenburg,[c] Jan Michels,[d] Ute Hentschel,[a,e] Beate M. Slaby[a]

aGEOMAR Helmholtz Centre for Ocean Research Kiel, RD3 Marine Symbioses, Kiel, Germany
bDepartment of Fisheries and Oceans, Bedford Institute of Oceanography, Dartmouth, Nova Scotia, Canada
cInstitute for Clinical Molecular Biology, Kiel University, Kiel, Germany
dFunctional Morphology and Biomechanics, Institute of Zoology, Kiel University, Kiel, Germany
eKiel University, Kiel, Germany

Kristina Bayer and Kathrin Busch are co-first authors and contributed equally to the manuscript. As the main focus lies on genomic interpretation, Kristina Bayer is named first.

**ABSTRACT** Few studies have explored the microbiomes of glass sponges (Hexactinellida). The present study seeks to elucidate the composition of the microbiota associated with the glass sponge *Vazella pourtalesii* and the functional strategies of the main symbionts. We combined microscopic approaches with metagenome-guided microbial genome reconstruction and amplicon community profiling toward this goal. Microscopic imaging revealed that the host and microbial cells appeared within dense biomass patches that are presumably syncytial tissue aggregates. Based on abundances in amplicon libraries and metagenomic data, SAR324 bacteria, *Crenarchaeota*, Patescibacteria, and *Nanoarchaeota* were identified as abundant members of the *V. pourtalesii* microbiome; thus, their genomic potentials were analyzed in detail. A general pattern emerged in that the *V. pourtalesii* symbionts had very small genome sizes, in the range of 0.5 to 2.2 Mb, and low GC contents, even below those of seawater relatives. Based on functional analyses of metagenome-assembled genomes (MAGs), we propose two major microbial strategies: the "givers," namely, *Crenarchaeota* and SAR324, heterotrophs and facultative anaerobes, produce and partly secrete all required amino acids and vitamins. The "takers," *Nanoarchaeota* and Patescibacteria, are anaerobes with reduced genomes that tap into the microbial community for resources, e.g., lipids and DNA, likely using pilus-like structures. We posit that the existence of microbial cells in sponge syncytia together with the low-oxygen conditions in the seawater environment are factors that shape the unique compositional and functional properties of the microbial community associated with *V. pourtalesii*.

**IMPORTANCE** We investigated the microbial community of *V. pourtalesii* that forms globally unique, monospecific sponge grounds under low-oxygen conditions on the Scotian Shelf, where it plays a key role in its vulnerable ecosystem. The microbial community was found to be concentrated within biomass patches and is dominated by small cells (<1 μm). MAG analyses showed consistently small genome sizes and low GC contents, which is unusual compared to known sponge symbionts. These properties, as well as the (facultatively) anaerobic metabolism and a high degree of interdependence between the dominant symbionts regarding amino acid and vitamin synthesis, are likely adaptations to the unique conditions within the syncytial tissue of their hexactinellid host and the low-oxygen environment.

**KEYWORDS** glass sponge, Porifera, Hexactinellida, symbiosis, microbiome, microbial metabolism, metagenomic binning, SAR324, *Crenarchaeota*, Patescibacteria, *Nanoarchaeota*, glass sponge

Address correspondence to Beate M. Slaby, bslaby@geomar.de.

Symbionts of the glass sponge Vazella pourtalesii with unusually small, low-GC genomes show different strategies for survival in the holobiont community

The fossil record shows that sponges (Porifera) have been essential members of reef communities in various phases of Earth's history and even built biohermal reefs in the mid-Jurassic to early-Cretaceous (1). Today, extensive sponge aggregations, also known as sponge grounds, are found throughout the World's oceans, from temperate to arctic regions along shelves, on ridges, and on seamounts (2). They can be mono- to multispecific with a single or various sponge species dominating the benthic community, respectively. In sponge ground ecosystems, these basal animals play a crucial role in the provision of habitat, adding structural complexity to the environment and thereby attracting other organisms, ultimately causing an enhancement of local biodiversity (3–5).

Studies on demosponges have shown that they harbor distinct and diverse microbial communities that interact with each other, their host, and the environment in various ways (6, 7). The microbial consortia of sponges are represented by diverse bacterial and archaeal communities, with ≥63 prokaryotic phyla having been found in sponges so far (6, 8). These sponge microbiomes display host species-specific patterns that are distinctly different from those of seawater in terms of richness, diversity, and community composition. Microbial symbionts contribute to holobiont metabolism (e.g., via nitrogen cycling and vitamin production) and defense (e.g., via secondary metabolite production) (reviewed in reference 7). Sponges and their associated microbial communities (here termed holobionts) further contribute to fundamental biogeochemical cycles like nitrogen, phosphorous, and dissolved organic matter in the ecosystem, but the relative contribution of microbial symbionts remains mostly unresolved (1, 7, 9, 10).

Sponges of the class Hexactinellida (glass sponges) are largely present and abundant in the mesopelagic realm below 400 feet. They can form extensive reefs of biohermal character and can dominate the sponge ground ecosystems (1, 11). Glass sponges are characterized by a skeleton of siliceous spicules that is six-rayed symmetrical with square axial proteinaceous filament (12). Much of the body is composed of syncytial tissue, which represents extensive and continuous regions of multinucleated cytoplasm (12, 13). Nutrients also are transported via the cytoplasmic streams of these trabecular syncytia (12). Some discrete cell types exist, including choanocytes and the pluripotent archaeocytes that are likely nonmotile and, thus, not involved in nutrient transport in Hexactinellida (12). While the microbial symbiont diversity and functions are well studied in Demospongiae, much less is known about the presence and function of microbes in glass sponges. In fact, microorganisms have rarely been seen in glass sponges (12). However, a recent study of South Pacific sponge microbial communities has shown that general patterns seen previously in shallow-water sponge microbiomes, such as host specificity and low microbial abundance-high microbial abundance dichotomy, are generally applicable for these deep-sea sponge microbiomes as well, including those of glass sponges (14). Another study underlined the importance of ammonia-oxidizing archaea (family *Nitrosopumilaceae*, phylum *Thaumarchaeota*) in the deep-sea hexactinellid *Lophophysema eversa* using metagenomic data (15).

Here, we investigate the microbial community of *Vazella pourtalesii* (16), a glass sponge (class Hexactinellida) that forms globally unique, monospecific sponge grounds on the Scotian Shelf off Nova Scotia, Canada. This ecosystem is characterized by relatively warm and nutrient-rich water with low oxygen concentrations (1, 17, 18). While there have been a number of studies on the distribution, biomarkers, and possible functional roles of *V. pourtalesii* in the ecosystem (1, 17, 19), little has been published to date on its associated microbiota (20). According to phylogenetic and fossil studies, sponges (including glass sponges) originate from Neoproterozoic times when oxygen was limited (21). Moreover, laboratory experiments have shown that sponges can cope with low oxygen levels for extended periods of time (22–24). Due to the observed low-oxygen conditions at the sampling location, we explored whether the *V. pourtalesii* microbiome contains compositional as well as functional adaptations to

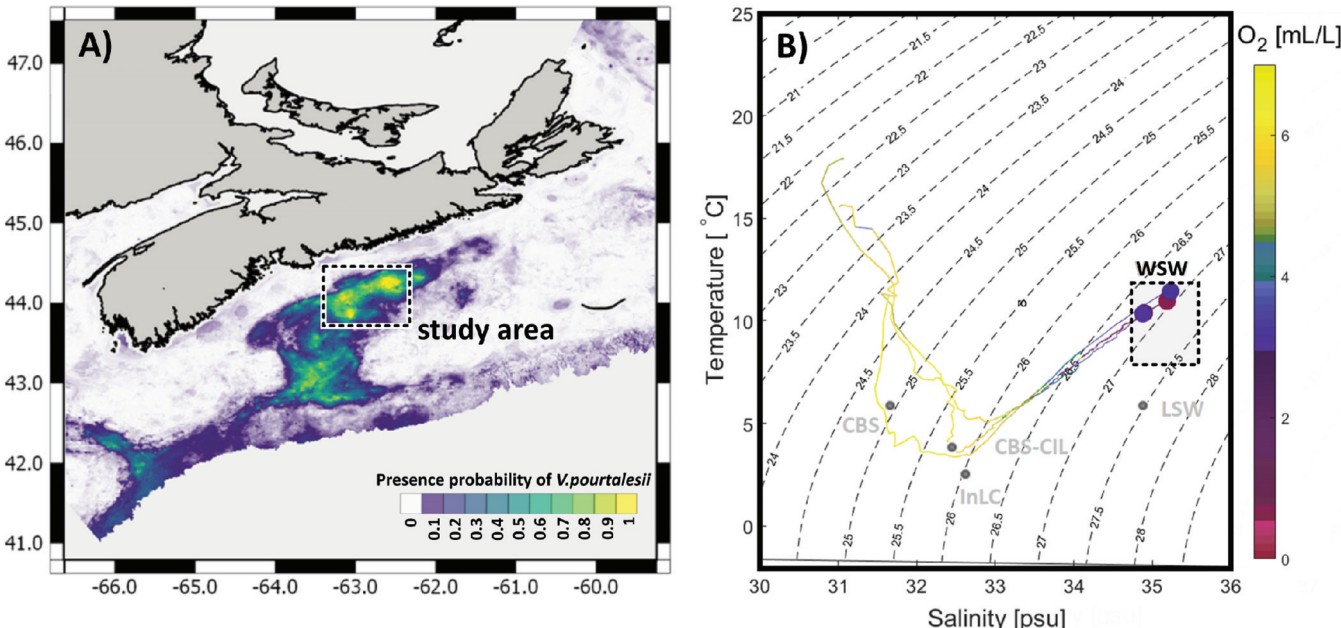

**FIG 1** Map of sampling region on the Canadian shelf (A) and TS diagram (B). (A) Colors depict presence probability of *Vazella pourtalesii* based on data presented in Beazley et al. (17), with yellow indicating areas of highest occurrence probability. (B) Coloring corresponds to oxygen concentrations measured during representative CTD casts at the study area. Water masses (light gray dots, labels, and square) were added according to Dever et al. (101) and Fratantoni et al. (102): CBS, Cabot Strait Subsurface Water; InLC, Inshore Labrador Current; CBS-CIL, Cold Intermediate Layer of Cabot Strait Subsurface Water; LSW, Labrador Slope Water; WSW, Warm Slope Water.

the low-oxygen environment. Microscopy, metagenome-guided microbial genome reconstruction, and amplicon community profiling were employed toward this goal.

## RESULTS

**Site description.** On the Scotian Shelf off Nova Scotia, eastern Canada, the highest densities of *V. pourtalesii* were observed and/or predicted in the Emerald Basin and the Sambro Bank areas (Fig. 1A). The water column of this region has a characteristic vertical structure, with water masses of different temperatures and salinities gradually mixing and creating a distinct temperature-salinity profile (Fig. 1B). Main water masses influencing the sampling sites are, from surface to deep sea, the following: Cabot Strait Subsurface Water (CBS), Inshore Labrador Current (InLC), Cold Intermediate Layer of Cabot Strait Subsurface Water (CBS-CIL), Labrador Slope Water (LSW), and Warm Slope Water (WSW). All *V. pourtalesii* samples of this study originate from a relatively warm ($>10°C$) and nutrient-rich water mass, called Warm Slope Water, which originates from the Gulf Stream (18). Relatively low oxygen concentrations ($<4$ mL/L) were measured at the sampling locations and depths that lay in the range of a mild hypoxia (25).

**Microscopic analyses of *V. pourtalesii*.** In contrast to other sponges, Hexactinellidae mainly consist of a single syncytium, a fusion of eukaryotic cells forming multinucleate tissue that permeates the whole sponge (12). By scanning electron microscopy (SEM), we observed that the overall amount of sponge biomass in *V. pourtalesii* was low, and its distribution within the spicule scaffolds was patchy (Fig. 2A and B). Closer inspection of such biomass patches by light microscopy and by transmission electron microscopy (TEM) (Fig. 2C and D) revealed numerous host cells with their characteristic nuclei, as well as high densities of microbial cells of various morphologies and with a dominance of comparatively small cell sizes ($<1$ $\mu$m). In addition, we frequently observed microbial cells that were attached to each other with pilus-like structures (Fig. 2E and F).

**MAG selection.** In total, 137 metagenome-assembled genomes (MAGs) of $>50\%$ estimated completeness and $<10\%$ redundancy were retrieved (see Table S1 in the supplemental material) from seven sponges and five seawater controls that were

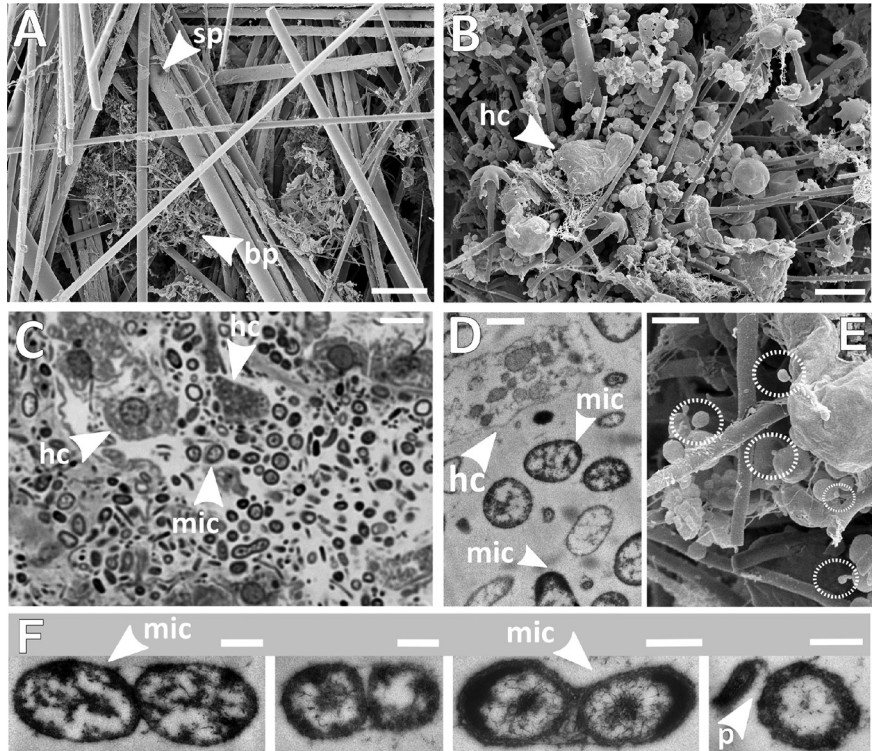

**FIG 2** Microscopy of *Vazella pourtalesii* tissue. (A) Scanning electron microscopy overview of spicule scaffolds (scale bar, 75 μm). (B) SEM closeup image of a biomass patch (scale bar, 3 μm). (C and D) Light microscopy image (scale bar, 5 μm) (C), and TEM image of the same biomass patch (scale bar, 1 μm) (D). (E) SEM closeup presumably showing smaller microbes attached to larger ones by stalk- or filament-like structures (scale bar, 1 μm). (F) TEM images of adjacent microbial cells (scale bars, 500 nm). Acronyms: sp, spicule; bp, biomass patch; hc, host cell; mic, microbes; p, potential pilus.

sampled from natural *Vazella* grounds and a mooring float (Table S2). Proteobacteria followed by Patescibacteria were the dominant bacterial phyla according to amplicon analyses. In addition, linear discriminant analysis (LDA) scores were obtained for MAGs based on their read abundance in the different metagenomic sample types (Fig. S1): (i) *V. pourtalesii* metagenomes versus seawater metagenomes and (ii) pristine *V. pourtalesii* versus mooring *V. pourtalesii* versus seawater metagenomes. Based on these assessments, we selected 13 representative *V. pourtalesii*-enriched MAGs of the four dominant phyla for detailed analyses (Table 1). Five MAGs belonged to the candidate phylum Patescibacteria, three to the candidate phylum SAR324, four to the phylum *Crenarchaeota*, and one to the phylum *Nanoarchaeota*. The selected MAGs are evidently not redundant representations of the same microbial genomes, which is visualized by the comparatively long branches in the phylogenomic tree (Fig. 3). Additionally, the MAGs showed a maximum of 86.8%, 93.2%, and 88.3% similarity to each other in the average nucleotide identity (ANI) analysis for SAR324, *Crenarchaeota*, and Patescibacteria, respectively (Table S4).

**Phylogenetic placement of the major players.** Three of the selected MAGs belong to the candidate phylum SAR324. The SAR324 clade was recently moved to the level of candidate phylum along with the publication of the whole-genome-based classification of microbial genomes (26). The three SAR324 MAGs clustered outside known orders with relatively long branches, showing that they are genomically distinct from published genomes of their closest relatives (Fig. 3). In the amplicon analysis, this taxon was placed within the class *Deltaproteobacteria*, in which it was the most abundant order (Fig. S2).

Three crenarchaeal MAGs clustered with *Cenarchaeum symbiosum* A (GCA000200715.1), associated with the sponge *Axinella mexicana* (27). One MAG (CrenArch_101) was

**TABLE 1** MAGs of bacterial candidate phyla Patescibacteria and SAR324 and archaeal phyla *Crenarchaeota* and *Nanoarchaeota*[a]

| Phylum-level affiliation and MAG | No. of contigs | Estimated genome size (Mb) | GC (%) | $N_{50}$ | Cov (%) | Red (%) |
|---|---|---|---|---|---|---|
| Patescibacteria | | | | | | |
| Patesci_129 | 109 | 0.71 | 31.8 | 5,408 | 67.37 | 0 |
| Patesci_30 | 95 | 0.46 | 31.7 | 5,089 | 66.19 | 1.18 |
| Patesci_136 | 206 | 0.95 | 31.6 | 4,840 | 61.21 | 0.47 |
| Patesci_48 | 163 | 0.86 | 32.6 | 8,405 | 71.52 | 0 |
| Patesci_98 | 15 | 0.71 | 28.4 | 66,386 | 79.44 | 0 |
| | | | | | | |
| SAR324 (*Deltaproteobacteria*) | | | | | | |
| SAR324_126 | 562 | 2.16 | 33.0 | 8,540 | 75.63 | 5.39 |
| SAR324_140 | 257 | 1.49 | 32.2 | 4,131 | 54.53 | 0.75 |
| SAR324_8 | 270 | 1.47 | 32.4 | 3,229 | 54.81 | 2.45 |
| | | | | | | |
| *Crenarchaeota* | | | | | | |
| CrenArch_143 | 507 | 2.04 | 38.2 | 3,080 | 63.44 | 1.46 |
| CrenArch_74 | 366 | 1.41 | 40.1 | 3,432 | 53.2 | 8.16 |
| CrenArch_90 | 115 | 1.11 | 40.4 | 8,261 | 59.77 | 2.59 |
| CrenArch_101 | 336 | 1.74 | 31.8 | 3,376 | 60.75 | 3.40 |
| | | | | | | |
| *Nanoarchaeota* | | | | | | |
| NanoArch_78 | 39 | 0.67 | 24.3 | 39,559 | 76.63 | 0 |

[a]Genome properties were determined by QUAST, and completeness and contamination estimations were performed by CheckM implemented in the metaWRAP pipeline. Acronyms: Cov, genome coverage; Red, redundancy.

closely related to *Nitrosopumilus* species isolated from Arctic marine sediment (28) and the deep-sea sponge *Neamphius huxleyi* (29). *Crenarchaeota* are not represented adequately in our amplicon data due to the sequencing primer bias toward bacteria.

Three of the five *V. pourtalesii*-enriched Patescibacteria MAGs clustered with genome GCA002747955.1 from the oral metagenome of a dolphin (30). This clade is a sister group to other families of the order UBA9983 in the class *Paceibacteria*. The other two patescibacterial MAGs belonged to the family *Kaiserbacteriaceae*. They were placed separate from each other and outside known genera, where they cluster with two different groundwater bacteria (GCA_000998045.1 and GCA_002773335.1). Patescibacteria were the second most abundant phylum in the amplicon data, with a dominance of the class *Parcubacteria* that showed high abundances in *V. pourtalesii* compared to seawater and sediment controls (Fig. S3). The majority of the Parcubacteria (73%) remain unclassified, while 27% are classified as order *Kaiserbacteria*. As this phylum has not been noticed as particularly abundant in sponge microbiomes before, we tested whether they are sponge specific by comparison with the reference database of the Sponge Microbiome Project (SMP) (8 and data not shown). The 899 patescibacterial *V. pourtalesii* ASVs matched to 42 SMP operational taxonomic units (OTUs) (mostly listed as "unclassified Bacteria" due to an older SILVA version). We identified three SMP OTUs matching *V. pourtalesii Kaiserbacteria* ASVs, namely, OTU0005080, OTU0007201, and OTU0159142. These OTUs occurred in 45, 26, and 7 sponge species, respectively, in the SMP reference database showing a global distribution.

The one *Nanoarchaeota* MAG, NanoArch_78, belongs phylogenomically to the class Aenigmarchaeia, where it is placed together with MAG GCA_002254545.1 from a deep-sea hydrothermal vent sediment metagenome and outside known families. Despite the primer bias toward bacteria in the amplicon analysis, the phylum *Nanoarchaeota* was among the most abundant microbial phyla in this analysis (Fig. S2).

**Genome sizes and GC contents.** With respect to genome size, the MAGs range from 0.46 Mb (Patesci_30) to 2.16 Mb (SAR324_126), including completeness values into the genome size estimations, with $N_{50}$ values between 3,080 (CrenArch_143) and 66,386 (Patesci_98) (Table 1). According to CheckM, between 53.2% and 79.4% of the genomes are covered and redundancies range from 0% to 8.16%. GC contents range from 24.3% (NanoArch_78) to 40.38% (CrenArch_90). We compared the MAGs to genomes of symbionts from other sponge species and of seawater-derived microbes of

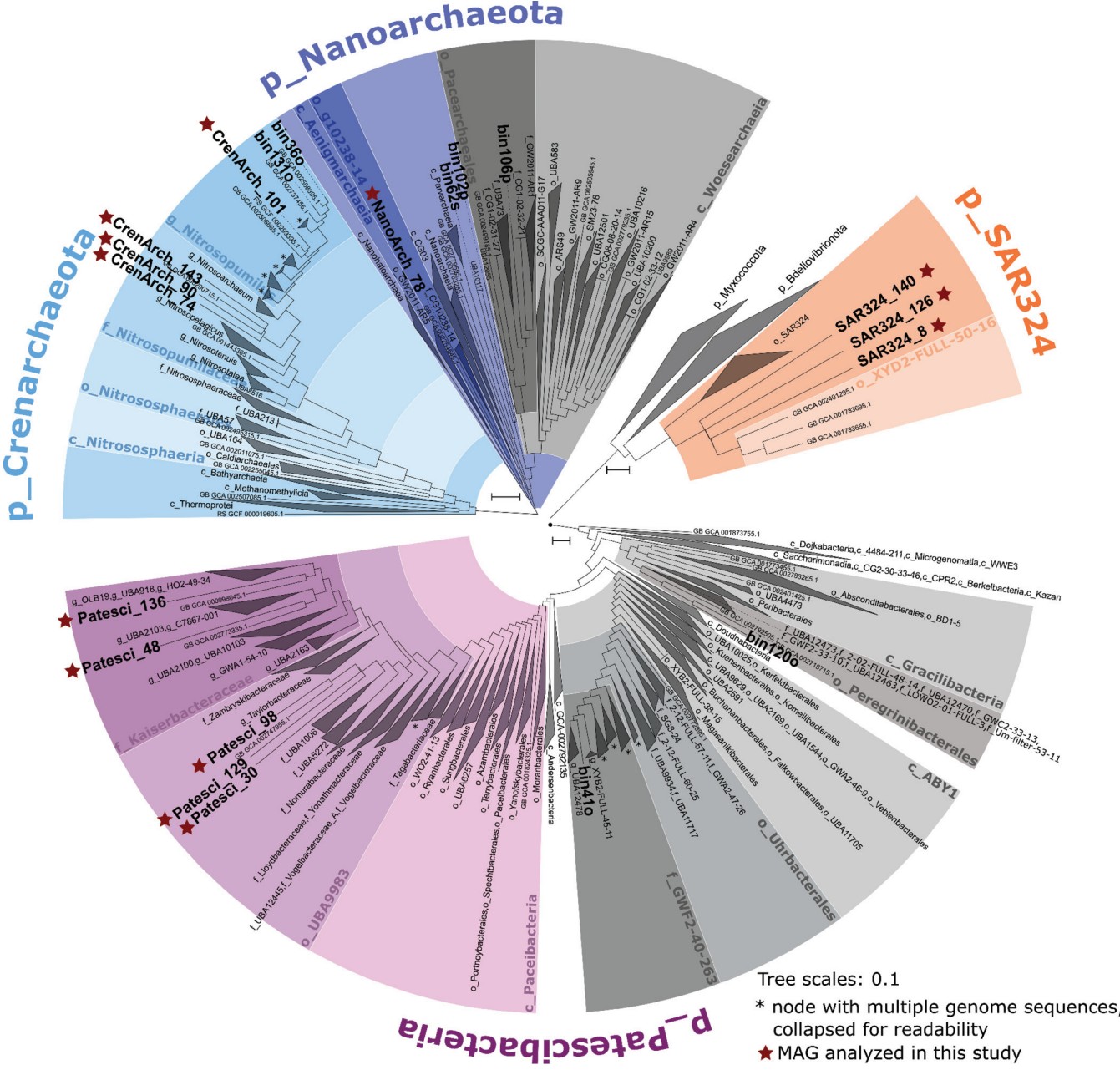

**FIG 3** Subtrees of the GTDB-Tk phylogenetic tree showing the setting of the MAGs of this study (shown in boldface) within the microbial phyla selected for detailed analysis. Class names are indicated by a leading "c_," order names by "o_," family names by "f_," and genus names by "g_." *Vazella*-enriched MAGs are marked with a red star.

each respective phylum regarding their estimated genome sizes and GC contents (Fig. 4 and Table S3). Due to the lack of published genomes of SAR324 and *Nanoarchaeota* sponge symbionts, we included genome size and GC content data of unpublished symbionts of *Phakellia* species and *Stryphnus fortis* in this analysis. This comparison revealed that the genomes of *V. pourtalesii*-enriched microbes are exceptionally small, with very low GC percentages. This trend is especially striking for Patescibacteria, SAR324, and *Nanoarchaeota*, as their values are not only low for sponge symbionts but even below the levels of the respective related seawater microbes. While this is not the case for *Crenarchaeota*, the *V. pourtalesii* MAGs are, nevertheless, in the lower ranges regarding size and GC content compared to other sponge symbionts.

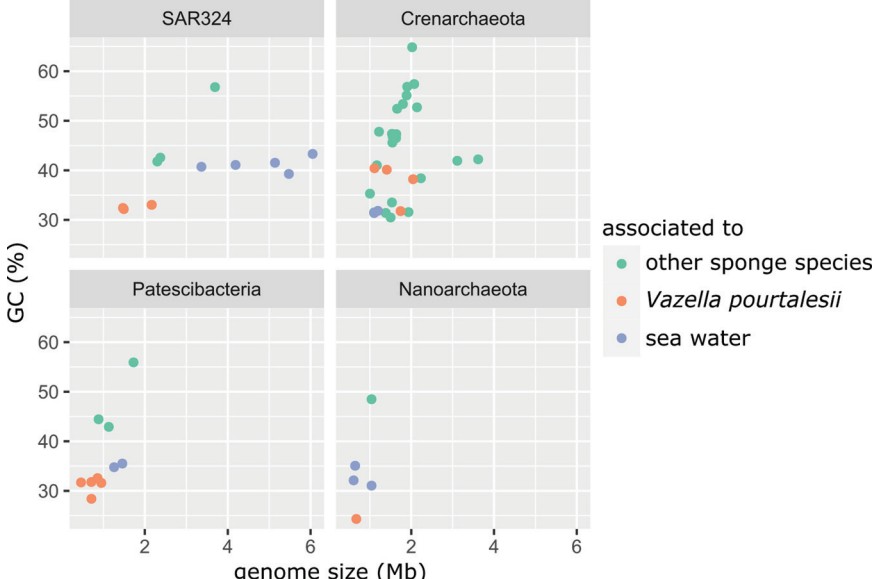

**FIG 4** Comparison of the MAGs retrieved in this study to published MAGs from sponge and seawater metagenomes. From this study, only the MAGs enriched in either *Vazella pourtalesii* or in water were considered; neutral ones were excluded from this analysis.

**Predicted lifestyle of the major players. (i) SAR324.** Metagenomic analysis of the three SAR324 MAGs from *V. pourtalesii* (Fig. 5A) revealed the presence of a nearly complete glycolysis pathway up to pyruvate (Pyr) along with the genes for the tricarboxylic acid (TCA) cycle and for conversions of the pentose phosphate pathway (PPP) (see Text S1 for in-depth analysis of more complex SAR324 and crenarchaeal MAGs). Pyruvate is converted aerobically by the pyruvate dehydrogenase enzyme complex into acetyl-coenzyme A (CoA), which fuels the completely annotated (thiamine-dependent) TCA cycle. While SAR324 have the genes for a nearly complete respiratory chain, their lifestyle appears to be facultatively anaerobic. We detected enzymes of the glyoxylate bypass (orange arrows within the TCA cycle in Fig. 5A), which is required by bacteria to grow anaerobically on fatty acids and acetate (31). This is supported by the presence of a potential AMP-dependent acetyl-CoA synthetase to utilize acetate, whereas enzymes for fatty acid degradation were not found. SAR324 might gain additional energy by a cation-driven p-type ATPase and possibly also anaerobic respiration (fumarate and nitrite/sulfide respiration) (also see Text S1). There is evidence for assimilatory sulfate reduction, but the pathway was not fully resolved.

V. pourtalesii-associated SAR324 organisms are able to take up di- and tricarboxylates using TRAP and TTT transporters, respectively. The imported substances can feed the TCA cycle under aerobic conditions or serve as an energy source through fumarate respiration under anaerobic conditions (32, 33). The presence of lactate dehydrogenase (LDH), involved in fermentation, further supports a facultatively anaerobic lifestyle (31).

SAR324 symbionts are capable of synthesizing diverse amino acids and B vitamins (riboflavin, coenzyme F420, folate, panthotheonate and thiamine from pyridoxal) using precursors from glycolysis, PPP, and the TCA cycle (summarized in Fig. 5A; see Text S1 for more details). Additionally, the genomes are well equipped with several transporters enabling the import and export of diverse substances (e.g., sugars, amino acids, peptides, and ions) (Text S1). These transporters are involved in, e.g., osmoregulation and/or toxic ion reduction (cobalt and fluoride) and multidrug resistance/import. A p-type ATPase was annotated that may aid in the export of cations or may use an electrochemical gradient for ATP synthesis. Proteins can be secreted by tat and sec transport as well as type 1 (T1SS) and type 2 (T2SS) secretion systems, probably involved in excretion of symbiosis-relevant molecules. We further identified autoinducer-2 (AI-2) and DNA-T family transporter (Text S1).

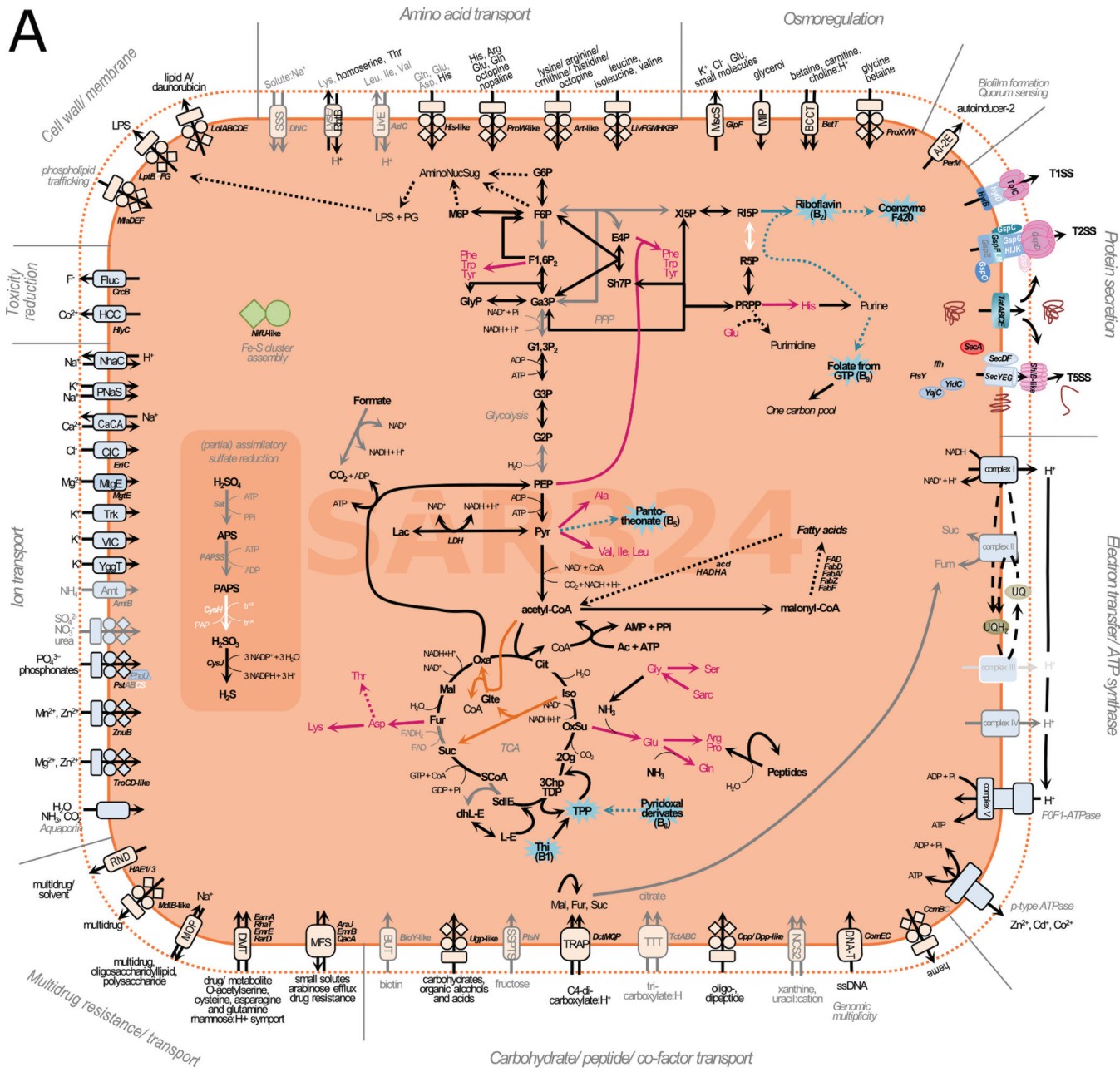

**FIG 5** Reconstruction of metabolic features found in the genomes of SAR324 (A), *Crenarchaeota* (B), Patescibacteria (C), and *Nanoarchaeota* (D). Solid lines indicate that genes/enzymes, or >50% of a given pathway, were found, and dashed lines indicate that <50% of a pathway was found. Gray arrows, writing, and lining indicate that the genes/enzymes were found in <50% of genomes of the respective phylum. White arrows and writing indicate missing genes/enzymes. Cofactor synthesis is indicated by turquoise color and amino acid production by magenta color. Symport, antiport, uniport, and direction are indicated by number and direction of arrows.

**(ii) Crenarchaeota.** The *Crenarchaeota* of *V. pourtalesii* (Fig. 5B) are capable of glycolysis from glucose-6-phosphate (G6P) to Pyr, which is likely anaerobically converted into acetyl-CoA using the enzyme pyruvate-ferredoxin oxidoreductase, which suggests facultatively anaerobic metabolism. The TCA cycle was almost completely annotated. Genomic evidence for aerobic and anaerobic respiration (fumarate and nitrite/sulfide respiration) was detected (Text S1). Genes for autotrophic $CO_2$ fixation in *V. pourtalesii*-associated microbes were lacking, but assimilatory sulfate reduction was annotated completely. A PPase was annotated that might deliver phosphates to feed the ATP synthase, and in one MAG, a transporter to import dicarboxylates (fumarate,

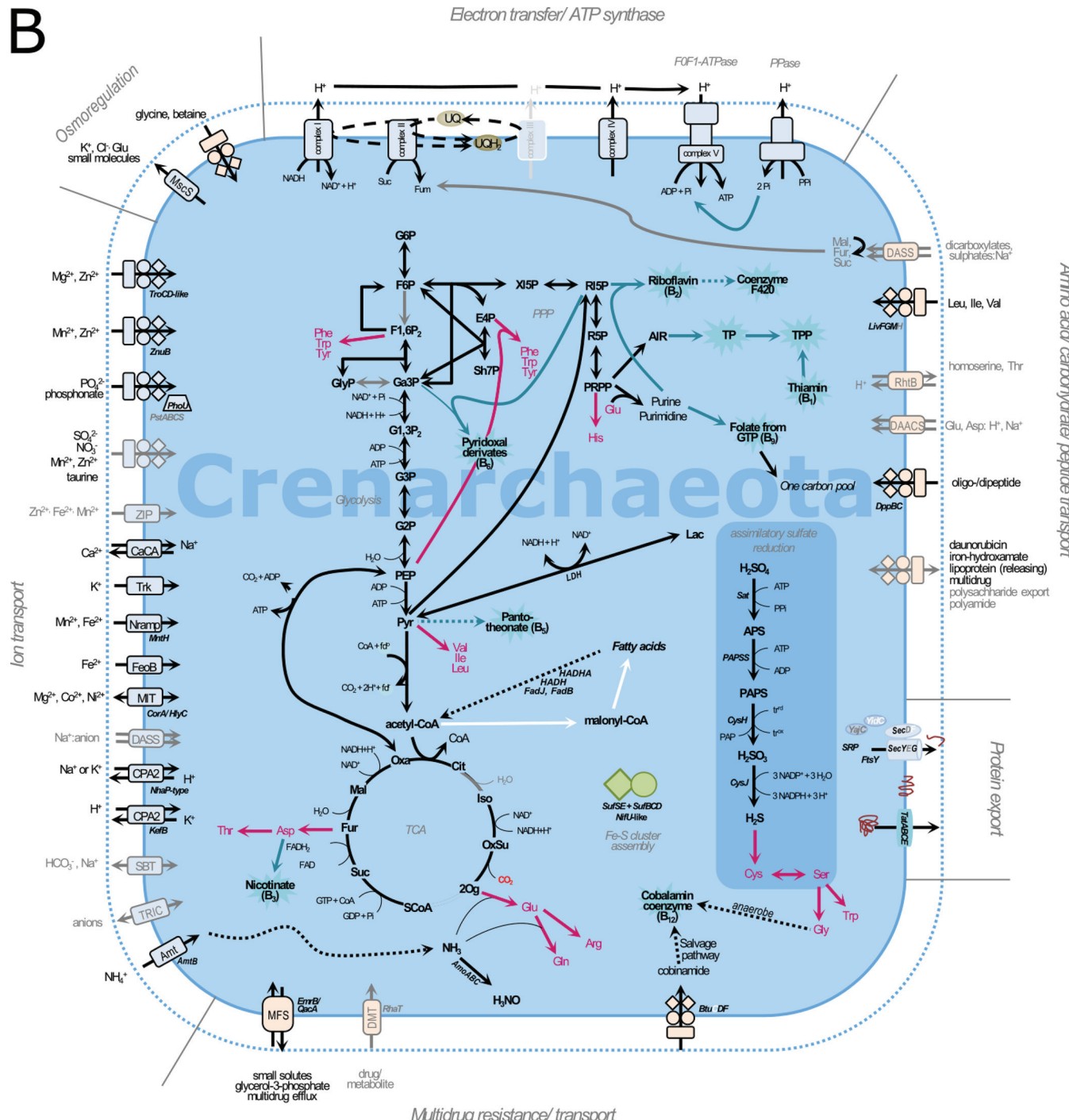

**FIG 5** (Continued)

malate, and succinate) was annotated, a feature we found in SAR324 as well. These substances could feed the TCA cycle under aerobic conditions. Additionally supporting the hypothesis of a facultatively anaerobic lifestyle is the presence of the enzyme LDH, which is involved in fermentation (31). The crenarchaeal genomes encode the synthesis of an even greater number of B vitamins than the SAR324 genomes, including riboflavin, coenzyme F420, folate, pantothenate, pyridoxal, nicotinate, and cobalamin (anaerobically), using precursors of central metabolism. Interestingly, they can synthesize thiamine from 5-phosphoribosyl diphosphate (PRPP), while SAR324 partially en-

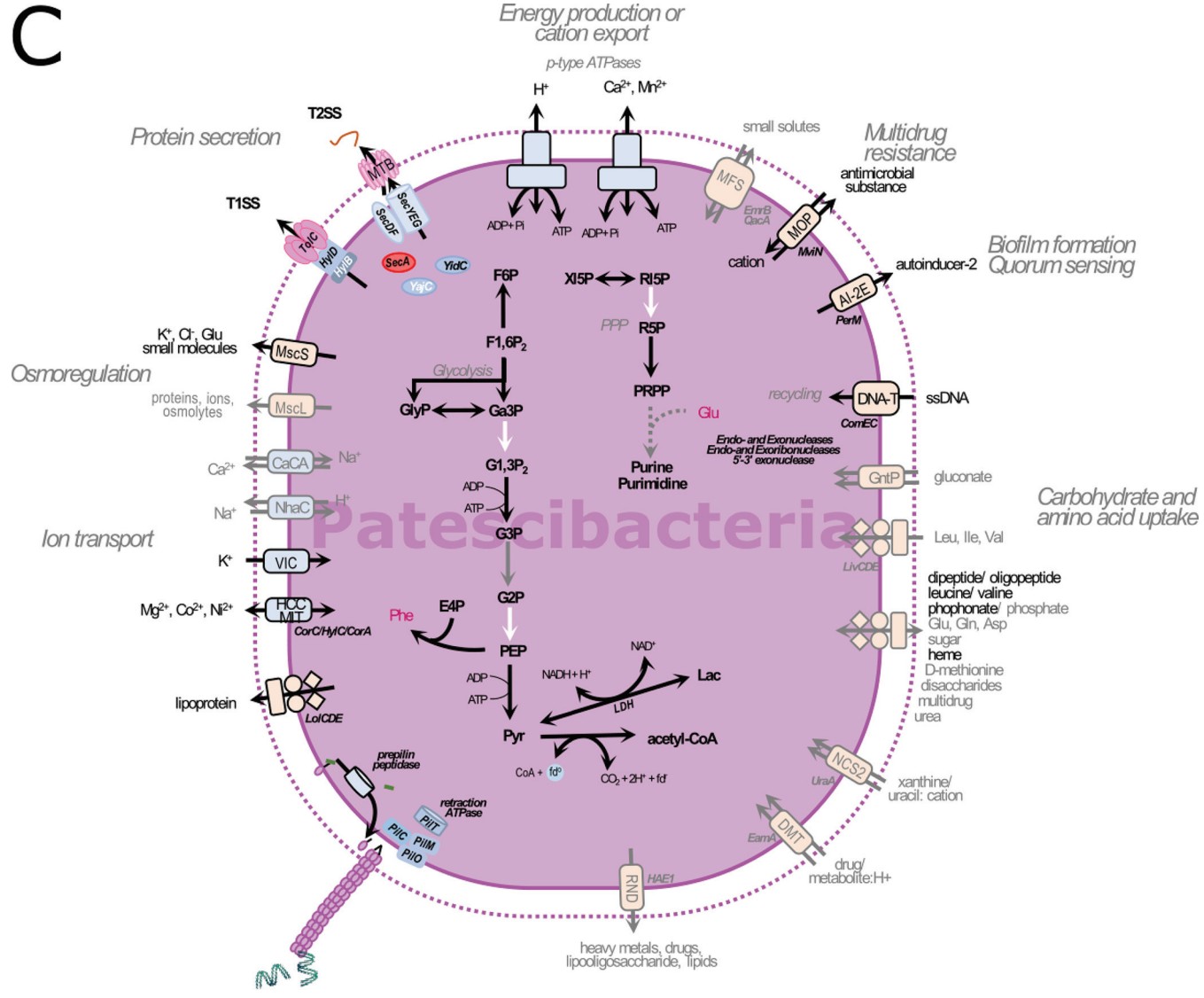

**FIG 5** (Continued)

code thiamine synthesis from pyridoxal, which would need to be imported from an external source (e.g., from other community members).

The crenarchaeal genomes are further well equipped with transporters that facilitate import and export of diverse substances, such as sugars, amino acids, peptides, and ions, among others (Text S1). These transporters are involved in, e.g., osmoregulation, reflecting the adaptation to a saline environment, and in multidrug resistance/import. A p-type ATPase was annotated in the genomes that may be involved in the export of cations (forced by ATP utilization). Protein secretion can be realized by tat and sec transport, which might be involved in the transport of proteins, such as those necessary for membrane formation and maintenance (Text S1).

**(iii) Patescibacteria.** The *V. pourtalesii*-associated Patescibacteria (Fig. 5C) showed metabolic capacity similar to that of published Patescibacteria from other environments. While we found several enzymes involved in glycolysis, we could not resolve the pathway completely. The genomes lack enzymes involved in oxidative phosphorylation (respiration) and the TCA cycle. The synthesis pathways of the important precursor PRPP and subsequent synthesis of purines and pyrimidines were only partially encoded. The biosynthesis of phenylalanine (Phe) from phosphoenolpyruvate (PEP) and

D

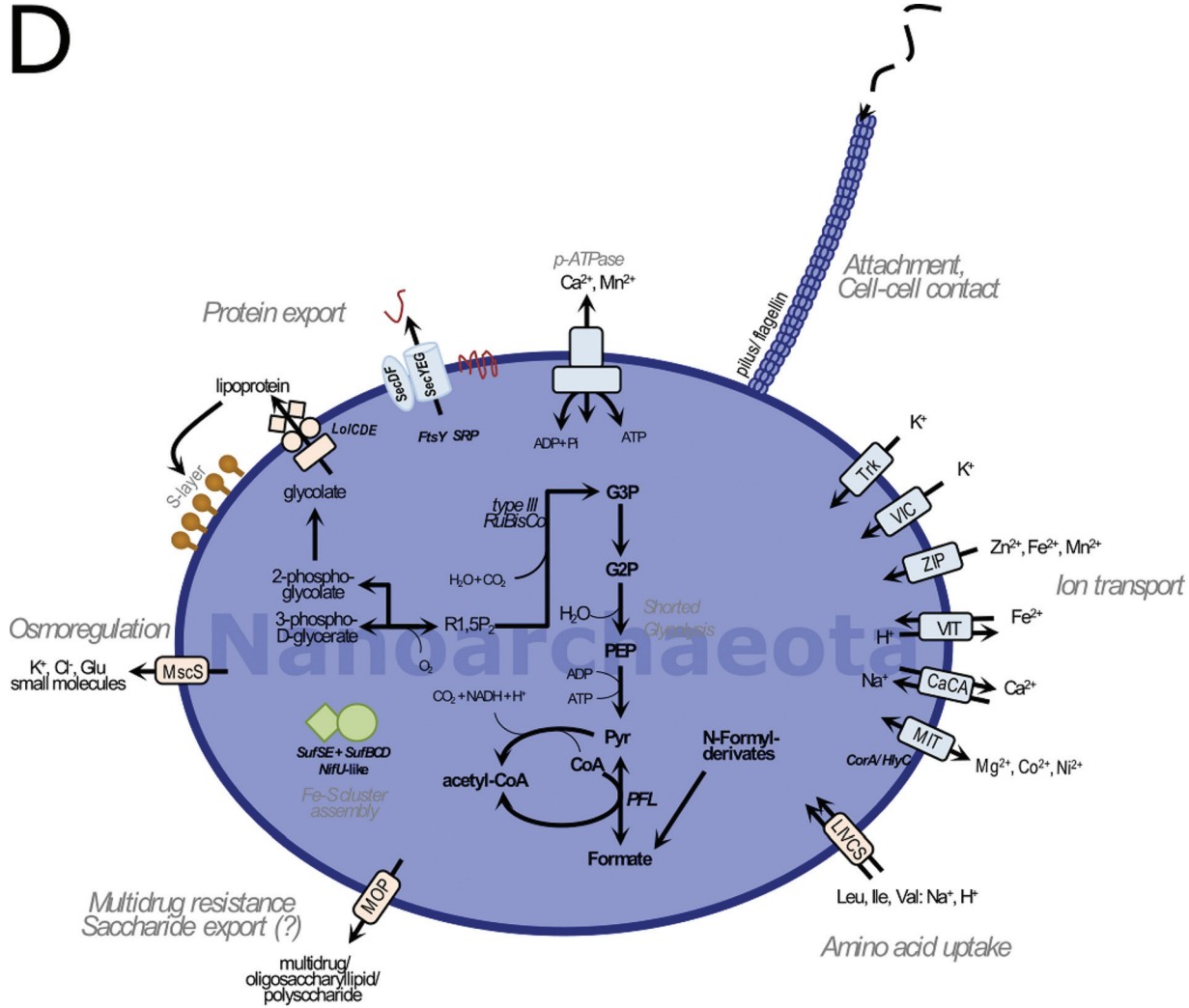

**FIG 5** (Continued)

erythrose-4-phosphate (E4P) is encoded, but biosynthesis pathways for other amino acids, cofactors, or vitamins are missing. We found two p-type ATPases that might export ions or provide energy (ATP) using cation and/or proton gradients present in the environment (holobiont). An anaerobic lifestyle is likely due to the presence of an LDH, an enzyme involved in lactate fermentation and anaerobic acetyl-CoA synthesis using the enzyme pyruvate-ferredoxin oxidoreductase. Patescibacterial MAGs also encode some transporters, albeit in lower numbers than the above-described SAR324 and crenarchaeal genomes. These transporters may be involved in osmoregulation, multidrug influx and efflux, and sugar, amino acid, and ion uptake. Regarding further symbiosis-relevant features, we detected the autoinducer transporter AI-2E. Additionally, we detected *ComEC/Rec2* and related proteins that are involved in the uptake of single-stranded DNA (34). This is supported by the presence of *PilT*, the motor protein that is thought to drive pilin retraction prior to DNA uptake, and the pilus assembly proteins *PilM*, *PilO*, and *PilC*. Even though the machinery was not annotated completely, Patescibacteria may be able to take up foreign DNA via retraction of type IV-pilin-like structures into the periplasm and via *ComEC* through the inner membrane.

**(iv) Nanoarchaeota.** The nanoarchaeal MAG (Fig. 5D) shows the genomic potential to convert glycerate-3-phosphate (G3P) to Pyr, which represents a shortened glycolysis pathway and results in reduced potential for energy production (ATP synthesis). It could use an anaerobic pyruvate-formate lyase (PFL) for acetyl-CoA production. Interestingly,

an archaeal type III RuBisCo is encoded that catalyzes light-independent $CO_2$ fixation using ribulose-1,5-bisphosphate and $CO_2$ as substrates to synthesize G3P (35) to fill the only partially encoded glycolysis. All enzymes needed for respiration were absent, supporting an anaerobic lifestyle. Energy might be gained using a cation-driven p-type ATPase. Like its published relatives (36, 37), the *V. pourtalesii*-associated nanoarchaeon lacks almost all known genes required for the *de novo* biosynthesis of amino acids, vitamins, nucleotides, cofactors, and lipids. The uptake of some amino acids and ions may be possible, as few transporters were detected. Typical archaeal S-layer membrane proteins are encoded that may be exported by an ABC-transporter (*LolCDE*) or by sec transport. Other transporters are involved in osmoregulation and in multidrug resistance and/or transport. We found a prepilin type IV leader peptidase encoded in the genome that is synthesized as a precursor before flagellin/pilin is incorporated into a filament (38).

## DISCUSSION

**Microbial associations in *V. pourtalesii* syncytia.** The glass sponge *V. pourtalesii* consists of a scaffold of spicules with cellular biomass concentrated in biomass patches that contain sponge as well as microbial cells (Fig. 2). We assume that the biomass patches were probably formed by dehydration of syncytial tissues during fixation, resulting in higher biomass densities than in the *in vivo* situation (12). Surprisingly high numbers of microbial cells were found within the observed biomass patches, considering that microbes have rarely been noticed in glass sponges previously (12). In *V. pourtalesii*, the microbes appeared in various morphotypes, indicating a taxonomically diverse microbial community. Microbial cells of strikingly small sizes ($<1$ $\mu$m) compared to those of shallow water demosponges (39) constituted a large fraction of the microbial community. Microbial cells were frequently seen in close association and even physically attached to each other (Fig. 2E and F). These associations were observed between equally sized cells but also between cells of distinctly different sizes, where the smaller microbes were attached to larger ones by stalk- or filament-like structures (Fig. 2E).

**Main players in the *V. pourtalesii* microbial community with small, low-GC genomes.** While previously published sponge metagenomes and MAGs tended toward high-GC contents (typically around 65 to 70%), the *V. pourtalesii* MAGs show lower GC levels in the range of 24 to 40% that are more similar to those of seawater metagenomes (40–42) (Fig. 4; see also Table S3 in the supplemental material). Genome sizes are also on the smaller side compared to previously published sponge MAGs (40, 42). The large genome sizes of demosponge symbionts may be attributed to the specific genomic toolbox they require to utilize the mesohyl matrix, such as carbohydrate-active enzymes (CAZy) and arylsulfatases. These genes are frequently found enriched in the sponge symbiont genomes compared to free-living relatives (42, 43). This is, however, not the case in *V. pourtalesii* MAGs. On the contrary, here we see GC contents and genome sizes similar to and even below the ones of free-living marine microbes of the same respective phyla (Fig. 4). Trophic specialization and avoidance of DNA replication cost have been proposed as hypotheses for genome reduction in free-living marine bacteria of, e.g., the genera *Idiomarina* (44) and *Pelagibacter* (45). For the *V. pourtalesii*-associated microbial community, the Black Queen hypothesis may best explain the apparent genomic streamlining: if some members carry out tasks that are beneficial to the whole microbial community, most other members will lose the ability to carry out these (often costly) tasks (46). Thus, the small sizes and low GC contents of the *V. pourtalesii* MAGs could be a sign of adaptation to generally nutrient-limiting environmental conditions (reviewed in reference 47) and specialization on nutrient sources that are available within the sponge host environment, such as ammonia.

**The "givers" and "takers" hypothesis.** Formerly placed within the *Thaumarchaeota*, the *Crenarchaeota* are well known and widespread sponge symbionts (48–51). Different genera have recently been observed in South Pacific Hexactinellida and Demospongiae (14). *Nanoarchaeota* and Patescibacteria also have recently been no-

ticed as members of sponge microbial communities, including glass sponges, but no sponge-associated nanoarchaeal genome has been studied to date, and, likely due to their low abundance in other sponge species, patescibacterial genomes have not been studied in detail (14, 52, 53). No sponge-derived genomes are available for the phylum SAR324 so far. The genomes of *V. pourtalesii* microbes lack a number of properties that we know from typical shallow-water sponge symbionts (7), such as the potential for the production of secondary metabolites and arylsulfatases, and they are not enriched in CAZy genes. This underlines the above-stated hypothesis that these sponge symbionts do not need a diverse toolbox of genes to make use of a complex mesohyl, like symbionts of Demospongiae. On the contrary, they seem to possess streamlined genomes to save resources and likely rely on each other for essential substances, such as certain amino acids and vitamins.

Based on the functional genetic content of the four microbial phyla that we analyzed in greater detail, we propose two major strategies: the givers, namely, SAR324 and *Crenarchaeota*, with comparatively larger, more complex genomes and likely bigger in cell size, and the takers, Patescibacteria and *Nanoarchaeota*, with reduced genomes and likely smaller cell sizes. We posit that the givers, being genomically well equipped, could be producing and partly secreting all required amino acids and vitamins, drawing energy from various aerobic as well as anaerobic processes (Fig. 6). Regarding their metabolic repertoire, the *Crenarchaeota* described here are rather similar to the SAR324 bacteria, namely, in their facultatively anaerobic lifestyle, the reactions of the central metabolism, and their ability for amino acid and vitamin biosynthesis. At the same time, while published sponge-associated or free-living *Crenarchaeota* have the genomic repertoire to fix carbon (37, 50, 54–56), such pathways appear to be absent from *V. pourtalesii*-associated *Crenarchaeota*. These findings indicate that the microbes are specifically adapted to the conditions within their respective host sponge and the surrounding environment, e.g., low-oxygen conditions in this case.

Supporting our hypothesis of genome streamlining in the sense of the Black Queen hypothesis, the two givers also seem to depend on each other metabolically: *Crenarchaeota* can produce several B vitamins, which might be used by members of SAR324. One example is pyridoxal, provided by *Crenarchaeota* to SAR324, which would be able to produce thiamine. Their genomic similarity, their difference from close relatives from other environments, and their metabolic interdependence reinforces our hypothesis that they are, in fact, symbionts specifically adapted to life within their *V. pourtalesii* host. Beyond the scope of the microbial community, microbial vitamin production may also have an important role in animal host metabolism, such as the respiratory chain, the synthesis of coenzyme A, proteins, fatty acids, nucleic acids, cofactor synthesis, and carbohydrate metabolism (Fig. 6). As previously hypothesized for Demospongiae symbionts (7), the capacity for vitamin synthesis by microbes associated with *V. pourtalesii* might be an important factor in maintaining the symbiosis with the animal host.

It is tempting to speculate that the takers, reduced in size and functional potential, would scavenge from their neighbors. Marine Patescibacteria are known (and named) for their reduced genomes and metabolic capacities (57–59), and *Nanoarchaeota* are known for their dependence on a crenarchaeal host, although in very different marine environments, such as hydrothermal vents (60, 61). Regarding the exchange of substances between microbes, we propose that the nanoarchaeal symbionts "ride" on the crenarchaeal symbionts, analogous to what is described for *Nanoarchaeum equitans* and *Ignicoccus hospitalis* (37, 62), which use pili for attachment and possibly also for metabolite uptake. Jahn et al. (62) showed that *Nanoarchaeum equitans* may get large amounts of lipids (and possibly other substances) from its associated *Ignicoccus hospitalis*. We hypothesize that the nanoarchaeum in *V. pourtalesii* is likewise directly associated with the abundant *Crenarchaeota* and receives, e.g., lipids and DNA via cell attachment and using pilus-like structures to maintain cell-cell contact (see, e.g., small cells attached to larger cells in Fig. 2E) (37, 60, 63). In turn, they might provide carbon to the microbial consortium by anaerobic $CO_2$ fixation. Patescibacteria could take up required nutrients from their microenvironment, also utilizing pili equipped with a *pilT*

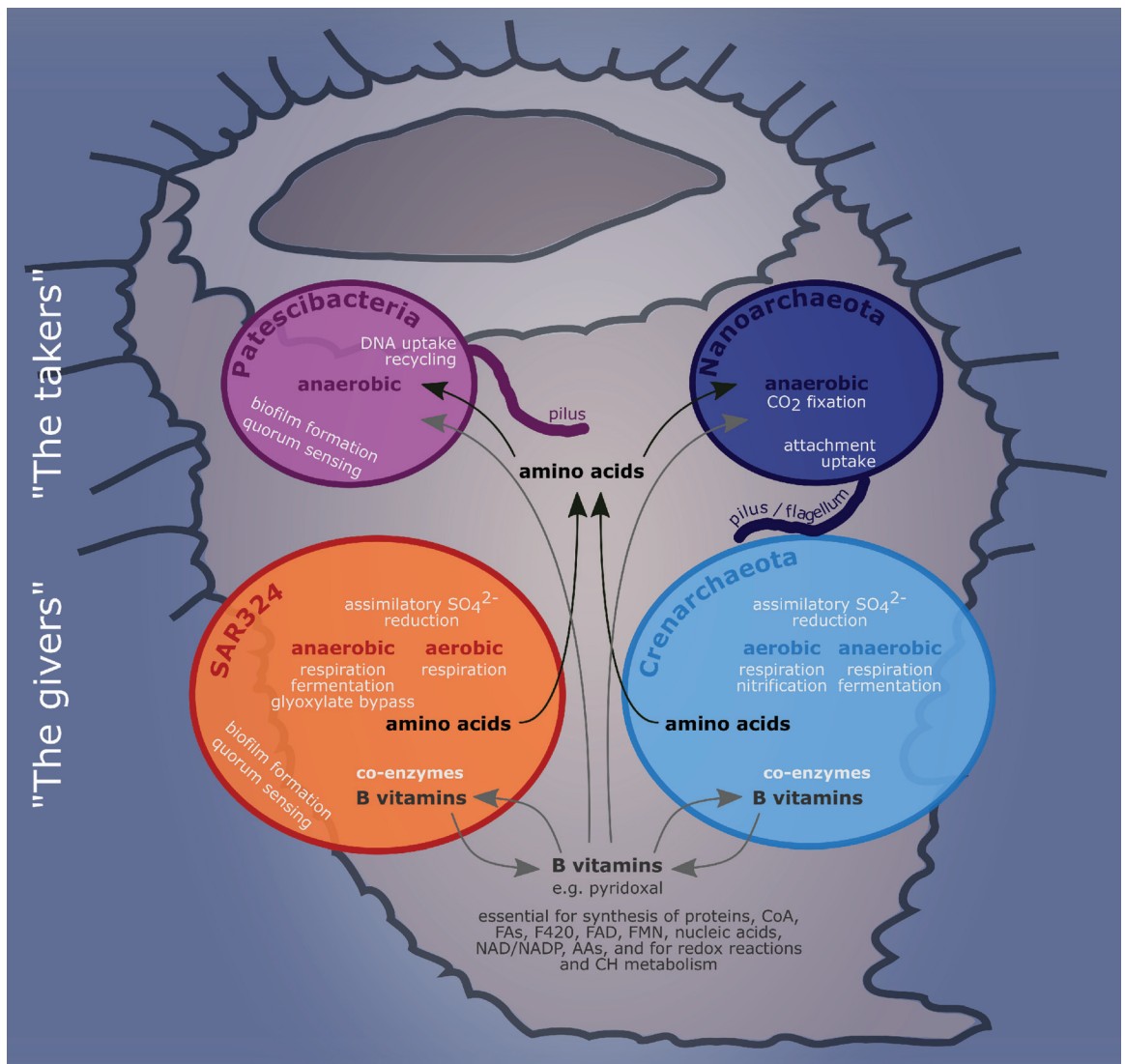

**FIG 6** Summary model of the main metabolic interactions between the four microbial taxa studied here. Black arrows indicate putative amino acid transport, gray arrows indicate B vitamin transport. CoA, acetyl-CoA; FAs, fatty acids; F420, coenzyme F420; FAD, flavin adenine dinucleotide; FMN, flavin mononucleotide; AAs, amino acids; CH, carbohydrate. The shape in the background depicts the host sponge *V. pourtalesii*.

motor protein and *comEC* enabling the uptake of DNA for the recycling of nucleotides (64) that they cannot build themselves.

Interestingly, we detected copies of the *luxS* gene, the proposed AI-2 synthetase, in patescibacterial and SAR324 genomes. There is strong evidence that AI-2E family homologues function as an AI-2 exporter in *E. coli* cells to control biofilm formation. AI-2 is a proposed signaling molecule for interspecies communication in bacteria (reviewed in reference 65). Autoinducer production plays a crucial role in *Vibrio fischeri* colonization of (and maintenance in) the light organ of the host squid *Euprymna scolopes* (66) and was recently detected in sponge-associated *Vibrio* species (67). Microbial quorum-sensing processes (such as biofilm formation, bioluminescence, motility, virulence factor secretion, antibiotic production, sporulation, and competence for DNA uptake) (68) may display symbiosis-relevant features additional or alternative to the ones described before for sponge-associated microbes (e.g., arylsulfatases, TPRs, and CRISPR-Cas).

**Conclusions.** The present study aimed to characterize the diversity and function of microbes residing in the glass sponge *Vazella pourtalesii*. A general pattern emerged in that the *V. pourtalesii* symbionts displayed smaller genome sizes and lower GC contents

than bacterial relatives from seawater or from demosponge symbionts. Genomic analysis revealed two putative functional strategies: the givers (SAR324 and *Crenarchaeota*) producing and most likely providing required amino acids and vitamins to the microbial community and the takers (Patescibacteria and *Nanoarchaeota*) depending on the provision of compounds like lipids and DNA that they likely take up via pilus-like structures. Their localization within biomass patches together with the environmental low-oxygen conditions could serve as an explanation for the unique compositional and functional properties of the microbial community of *V. pourtalesii*.

## MATERIALS AND METHODS

**Sampling and assessment of microbial community composition.** Sampling was performed on a cruise to the Scotian Shelf off Nova Scotia, eastern Canada, in August-September 2017 onboard CCGS *Martha L. Black* (MLB2017001). Here, we selected a subset of all samples received during this cruise (see Table S2 in the supplemental material) to study the lifestyle strategies of the dominant members of the microbial community. For details on sampling of sponges as well as seawater and sediment controls, DNA extraction, and amplicon sequencing, see reference 20, where we cover the complete data set to study the microbial diversity inside *V. pourtalesii* in response to anthropogenic activities. Briefly, sponge individuals were collected for this study from pristine areas by the remotely operated vehicle (ROV) ROPOS (Canadian Scientific Submersible Facility, Victoria, Canada), and tissue subsamples were taken, rinsed in sterile filtered seawater, and frozen at −80°C. Samples were collected at an average sampling depth of 168 m (minimum, 161 m; maximum, 183 m), which coincides with the base of the euphotic zone. Oceanographic data, such as temperature, salinity, and oxygen, were collected using conductivity, temperature, and depth (CTD) rosette casts (sensors by Sea-Bird Electronics SBE 25). Water samples were taken during CTD casts and using Niskin bottles of the ROPOS ROV. Sediment samples were collected at the same location as sponge and seawater samples by ROV push corers (Table S2). Additionally, sponge samples were collected from an Ocean Tracking Network (OTN) acoustic mooring located approximately 10 km northwest of the Sambro Bank Sponge Conservation Area on the Scotian Shelf (20). The mooring was anchored ~5 m above the seabed and was deployed for ~13 months (15 August 2016 to 5 September 2017) prior to its recovery.

DNA was extracted using the DNeasy power soil kit (Qiagen). After quantification and quality assessment by NanoDrop spectrophotometer and by PCR, the V3V4 variable regions of the 16S rRNA gene were amplified in a dual-barcoding approach (69) using a one-step PCR with the primers 5'-CCTACGGGAGGCAGCAG-3' (70) and 5'-GGACTACHVGGGTWTCTAAT-3' (71). Samples were sequenced on a MiSeq platform (MiSeqFGx; Illumina) with v3 chemistry. The raw sequences were quality filtered using BBDUK (BBMAP version 37.75 [72]) with a Q20 and a minimum length of 250 nucleotides. Sequences were processed in QIIME2 (versions 2018.6 and 2018.8 [73]), implementing the DADA2 algorithm (74) to determine amplicon sequence variants (ASVs). Sequences were denoised, and chimeras, chloroplasts, and mitochondrial sequences were removed. Taxonomy was assigned using a naïve Bayes classifier (75) trained on the SILVA 132 99% OTUs 16S database (76). An ASV-based phylogeny was generated using the FastTree2 plugin (77). The plots were produced with R (version 3.0.2 [78]), Inkscape (version 0.92.3 [79]), QGIS (version 2.18.4 [80]), and MATLAB (version R2016b, including Gibbs Seawater toolbox [81]).

**Scanning electron microscopy.** Tissue subsamples of three sponge individuals (Table S2) were fixed for SEM onboard ship in 6.25% glutardialdehyde in phosphate-buffered saline (PBS) (Fisher Scientific) in two technical replicates each. Samples were then washed three times for 15 min each time in PBS, postfixed for 2 h in 2% osmium tetroxide (Carl Roth, Germany), and washed again three times for 15 min each time in PBS. Samples were dehydrated in an ascending ethanol (EtOH) series (ROTIPURAN; Carl Roth, Germany): 1× 15 min 30% EtOH, 2× 15 min 50% EtOH, 2× 15 min 70% EtOH, 2× 15 min 80% EtOH, 2× 15 min 90% EtOH, and 1× 15 min 100% EtOH. Subsequent dehydration was continued with carbon dioxide in a critical point dryer (BalzersCPD 030). After critical point drying, the samples were manually fractionated and sputter coated for 3 min at 25 mA with gold/palladium (Balzers SCD 004). The preparations were visualized using a Hitachi S-4800 field emission scanning electron microscope (Hitachi High-Technologies Corporation, Tokyo, Japan) with a combination of upper and lower detectors at an acceleration voltage of 3 kV and an emission current of 10 mA.

**Transmission electron microscopy and light microscopy.** Tissue samples of three sponge individuals (Table S2) were fixed onboard ship in 2.5% glutardialdehyde in 0.1 M natriumcacodylate buffer (pH 7.4; Science Services GmbH) for TEM and light microscopy in two technical replicates each. Samples were then rinsed 3× with buffer at 4°C, postfixed for 2 h in 2% osmium tetroxide (Carl Roth), and washed with buffer three times for 15 min each time at 4°C. Samples were partially dehydrated with an ascending ethanol (ROTIPURAN; Carl Roth) series (2× 15 min 30% EtOH, 1× 15 min 50% EtOH, up to 70% ethanol). Samples were stored at 4°C overnight before desilicification with 4% suprapure hydrofluoric acid (incubation of approximately 5 h; Merck). Afterwards, samples were washed eight times for 15 min each time in 70% EtOH (with an overnight storage at 4°C in between washings). Dehydration was continued with a graded ethanol series (1× 15 min 90% EtOH and 2× 15 min 100% EtOH) followed by gradual infiltration with LR-White resin (Agar Scientific) at room temperature (1× 1 h 2:1 ethanol:LR-White, 1× 1 h 1:1 ethanol:LR-White, 1× 1 h 1:2 ethanol:LR-White; 2× 2 h pure LR-White). Samples were incubated in pure LR-White resin at 4°C overnight before being transferred into fresh resin and polymerized in embedding capsules at 57°C for 2 days.

Semithin sections (0.5 $\mu$m) were cut (in technical replicates) using an ultramicrotome (Ultracut E; Reichert-Jung) equipped with a diamond knife (Diatome, Switzerland) and were stained with Richardson solution (ingredients from Carl Roth; prepared as described in reference 82). Semithin sections were then mounted on SuperFrost Ultra Plus microscopy slices (using Biomount medium produced by Plano; Carl Roth) and visualized with an Axio Observer.Z1 microscope (Zeiss, Germany). Ultrathin sections (70 nm) were cut (in technical replicates) with the same ultramicrotome, mounted on pioloform coated copper grids (75 mesh; Plano), and contrasted with uranyl acetate (Science Services; 20-min incubation followed by washing steps with MilliQ water) and Reynold's lead citrate (ingredients from Carl Roth; 3-min incubation followed by washing steps with MilliQ water). The ultrathin preparations were visualized at an acceleration voltage of 80 kV on a Tecnai G2 Spirit Bio Twin transmission electron microscope (FEI Company).

**Microbial functional repertoire.** For metagenomic sequencing, DNA was extracted from the seven sponge samples (four from natural *Vazella* grounds and three from the mooring to optimize for differential coverage binning) and five seawater controls (Table S2) with the Qiagen AllPrep DNA/RNA minikit. Two washing steps with buffer AW2 were employed during DNA extraction. DNase and protease-free RNase A (Thermo Scientific) were used to remove remnant RNA from the DNA extracts. For seawater controls, DNA was extracted from half of a polyvinylidene difluoride membrane filter (seawater filter; described above), while the other half of the filter was used for amplicon analyses. The DNA was concentrated by precipitation with 100% ethanol and sodium acetate buffer and reeluted in 50 $\mu$l water. For all extracts, DNA quantity and quality were assessed by NanoDrop measurements and Qubit assays, and 30 $\mu$l (diluted in water, if necessary) was sent for metagenomic Illumina Nextera sequencing (HiSeq 4000, 2× 150-bp paired ends) at the Institute of Clinical Molecular Biology (IKMB) of Kiel University. The sequence quality of all read files was assessed with FastQC (83).

The raw reads were trimmed with Trimmomatic v0.36 (ILLUMINACLIP:NexteraPE-Pe.fa:2:30:10 LEADING:3 TRAILING:3 SLIDINGWINDOW:4:15 MINLEN:36) and coassembled with megahit v1.1.3 (84). The metaWRAP v1.0.2 pipeline was implemented for binning as follows (85). Initial binning was performed with metabat, metabat2, and maxbin2 (86–88) within metaWRAP. The bins were refined with the metaWRAP bin_refinement module and further improved, where possible, with the reassemble_bins module. This module uses the genome assembler SPAdes v3.12.0 (89) on two sets of reads mapped to the original bin with strict and more permissive settings and then compares the original bin with the two newly assembled genomes. Which of the three versions of the MAG was the best in each respective case and, thus, was used for further analyses is indicated by the trailing letter in the names (o, original; p, permissive; s, strict) in Table S1. The MAGs that were further analyzed in detail were renamed, indicating their phylum-level affiliation and their bin number.

MAG taxonomy was determined by GTDB-Tk based on whole-genome information and by following the recently published, revised microbial taxonomy by Parks and colleagues (26, 90, 91). The phylogenomic trees produced by GTDB-Tk (77, 92, 93) were visualized on the Interactive Tree Of Life (iTOL) platform v4.3 (94). MAG abundance in the different metagenomic data sets was quantified with the metaWRAP quant_bins module and used to determine which MAGs were enriched in which sample type by calculating LDA scores with LEfSe v1.0 (95) in two ways: (i) "*V. pourtalesii*" versus "water" and (ii) "pristine *V. pourtalesii*" versus "mooring *V. pourtalesii*" versus "water." We identified MAGs belonging to the bacterial candidate phyla Patescibacteria and SAR324 and the archaeal phyla *Crenarchaeota* and *Nanoarchaeota* that were enriched in *V. pourtalesii* over seawater or in one of the *V. pourtalesii* subsets. The MAGs were compared to each other within their taxonomic groups using ANI of the pangenomic workflow of anvi'o v5.2 (96, 97), and they were compared to seawater and other host sponge-derived reference genomes (Table S3) regarding their genome sizes and GC contents. For functional annotations, interproscan v5.30-69.0, including GO term and pathway annotations, was used (98, 99). The resulting EC numbers were converted to K terms with an in-house R script using the KEGG Orthology (KO) reference hierarchy to apply the online tool Reconstruction Pathway in KEGG mapper (https://www.genome.jp/kegg/mapper.html). Additionally, manual search in the annotation tables (https://doi.org/10.6084/m9.figshare.12280313) allowed the identification of several enzymes completing some pathways. Potential transporters were identified in the above-described annotation and using the online tool TransportDB 2.0 (100).

**Data deposition.** Detailed sample metadata was deposited in the PANGAEA database (https://doi.pangaea.de/10.1594/PANGAEA.917599). Amplicon and metagenomic raw read data were deposited in the NCBI database under BioProject PRJNA613976. Individual accession numbers for assembled MAGs are listed in Table S1. Interpro annotation output is available on figshare at https://doi.org/10.6084/m9.figshare.12280313.

## SUPPLEMENTAL MATERIAL

Supplemental material is available online only.

**TEXT S1**, DOC file, 0.1 MB.
**FIG S1**, EPS file, 0.8 MB.
**FIG S2**, EPS file, 1 MB.
**FIG S3**, EPS file, 0.7 MB.
**TABLE S1**, XLSX file, 0.2 MB.
**TABLE S2**, XLSX file, 0.01 MB.
**TABLE S3**, XLSX file, 0.01 MB.
**TABLE S4**, XLSX file, 0.01 MB.

## ACKNOWLEDGMENTS

The project "SponGES" received funding from the European Union's Horizon 2020 research and innovation program under grant agreement no. 679849. We thank our colleague Hans Tore Rapp for leading this great consortium for the last 4 years; you are dearly missed. Ship time and Canadian participation were enabled by Fisheries and Oceans Canada's (DFO) International Governance Strategy Science Program through project "Marine Biological Diversity Beyond Areas of National Jurisdiction (BBNJ): 3-Tiers of Diversity (Genes-Species-Communities)," led by E.K. (2017 to 2019). We acknowledge financial support by Land Schleswig-Holstein within the funding programme Open Access Publikationsfonds.

We appreciated the onboard support of the crew and scientific party of the expedition *MLB2017001*. Fred Whoriskey provided the OTN mooring samples. Andrea Hethke, Ina Clefsen, and the CRC1182 Z3 team (Katja Cloppenborg-Schmidt, Malte Rühlemann, and John Baines) provided valuable support with the amplicon pipeline. Further, we thank the following people for microscopy-related support: Yu-Chen Wu, Marie Sieberns, Anke Bleyer, Cay Kruse, and Julia-Vanessa Böge. Martin Jahn provided advice for metabolic analyses.

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
