## [Reviewer comments · mSystems]

Microbial strategies for survival in the glass sponge *Vazella pourtalesii*

Kristina Bayer, Kathrin Busch, Ellen Kenchington, Lindsay Beazley, Sören Franzenburg, Jan Michels, Ute Hentschel, and Beate Slaby

Corresponding Author(s): Beate Slaby, GEOMAR Helmholtz Centre for Ocean Research Kiel

Review Timeline:

Submission Date:	May 27, 2020
Editorial Decision:	June 30, 2020
Revision Received:	July 20, 2020
Accepted:	July 23, 2020

Editor: Seth Bordenstein

Reviewer(s): Disclosure of reviewer identity is with reference to reviewer comments included in decision letter(s). The following individuals involved in review of your submission have agreed to reveal their identity: Zhiyong Li (Reviewer #1)

Transaction Report:

DOI: <https://doi.org/10.1128/mSystems.00473-20>

June 30, 2020

Dr. Beate M Slaby
GEOMAR Helmholtz Centre for Ocean Research Kiel
Marine Symbioses
Düsternbrooker Weg 20
Kiel 24105
Germany

Re: mSystems00473-20 (Microbial strategies for survival in the glass sponge *Vazella pourtalesii*)

Dear Dr. Beate M Slaby:

Thank you for submitting your manuscript to mSystems. I am pleased to inform you that your manuscript is on track for acceptance pending some revisions noted below by the two expert reviewers. mSystems generally prefers a minimal amount of supplementary material so during the course of your revision, please consider moving supplementary figures to the main text if you deem it possible.

To submit your modified manuscript, log onto the eJP submission site at <https://msystems.msubmit.net/cgi-bin/main.plex>. If you cannot remember your password, click the "Can't remember your password?" link and follow the instructions on the screen. Go to Author Tasks and click the appropriate manuscript title to begin the resubmission process. The information that you entered when you first submitted the paper will be displayed. Please update the information as necessary. Provide (1) point-by-point responses to the issues raised by the reviewers as file type "Response to Reviewers," not in your cover letter, and (2) a PDF file that indicates the changes from the original submission (by highlighting or underlining the changes) as file type "Marked Up Manuscript - For Review Only."

Due to the SARS-CoV-2 pandemic, our typical 60 day deadline for revisions will not be applied. I hope that you will be able to submit a revised manuscript soon, but want to reassure you that the journal will be flexible in terms of timing, particularly if experimental revisions are needed. When you are ready to resubmit, please know that our staff and Editors are working remotely and handling submissions without delay. If you do not wish to modify the manuscript and prefer to submit it to another journal, please notify me of your decision immediately so that the manuscript may be formally withdrawn from consideration by mSystems.

To avoid unnecessary delay in publication should your modified manuscript be accepted, it is important that all elements you upload meet the technical requirements for production. I strongly recommend that you check your digital images using the Rapid Inspector tool at <http://rapidinspector.cadmus.com/RapidInspector/zmw/>.

Sincerely,

Seth Bordenstein

Editor, mSystems

Journals Department
Reviewer comments:

Reviewer #1 (Comments for the Author):

The manuscript 'Microbial strategies for survival in the glass sponge *Vazella pourtalesii*' by Kristina Bayer¹ et al. investigated the composition of the microbiota associated with the glass sponge *Vazella pourtalesii* and the functional strategies of the main symbionts using combined strategy of microscopic approaches with metagenome-guided microbial genome reconstruction and amplicon community profiling. Based on metagenome-assembled genomes analyses, SAR324 bacteria, Crenarchaeota, Patescibacteria and Nanoarchaeota were identified as abundant members of the *V. pourtalesii* microbiome. 13 representative *V. pourtalesii*-enriched MAGs from 137 were selected for detailed metabolic features analyses. Particularly the metabolic interactions between the four microbial taxa were highlighted. It was suggested that Crenarchaeota and SAR324 produce and partly secrete all required amino acids and vitamins. Nanoarchaeota and Patescibacteria, tap into the microbial community for resources, e.g., lipids and DNA, likely using pili-like structures. The results expand our understanding of the interaction between uncultured sponge microbes in a sponge holobiont. It is recommended for publication after minor revision:

1. What are the roles of Nanoarchaeota and Patescibacteria? Only takers? It will strengthen the interaction between the microbes if some functional features of Nanoarchaeota and Patescibacteria were revealed.
2. The references format needs to be unified: The first letter of the title words should be lowercase except the first word. References 14, 30, 32,33. 44, 51,53,56, 58,61,64,66,,68,101,102..... are wrong. The format of book references are suggested to be adjusted according to the journal requirement, for example 1,9,12, 31.....

Reviewer #2 Comments:

The work from Kristina Bayer and coauthors entitled "Microbial strategies for survival in the glass

sponge *Vazella pourtalesii*" describes the microbial community of the glass sponge *vazella pourtalessi* using microscopic description, amplicon analysis and metagenome-assembled genomes (MAGs) from 7 samples of sponges. The metabolic capacities are described based in K terms and other resources, which is also linked to the microscopic results and amplicon relative abundances.

I find this work to be quite interesting and well executed. Information about glass sponges is more scarce than demosponges, therefore this is a valuable resource for the sponge community. The authors have done a great effort analysing the functional repertory of dominant sponge symbionts, providing a summary in the main text and a detailed description in supplementary material. The manuscript is well-written and Introduction and Discussion are very nicely presented.

Overall, I think this manuscript would be an excellent fit for mSystems, and with some revisions and pending some clarifications from the authors I would recommend acceptance. My main concern is the specific reason for the selected MAGs. The selection of 13 MAGs is the key point for this study, since everything else is based on those selected ones (Line 141). MAGs are chosen based somehow in their enrichment in sponge samples versus water (LDA scores), in their completeness, and redundancy values, however, I don't see a clear justification for selecting those ones and not others.

For instance, why was bin_27 (the forth one with highest LDA score, 87.9 completeness, and being alphaproteobacteria) not selected but it was bin_136s with 61.2 completeness and lower LDA score? Is there a cutoff of some kind? Is it based on the taxonomy of the MAGs? Was this a personal selection? I wonder if the selection of other MAGs would bring a completely different story.

As a general comment, I want to point out that in this journal Methods are presented at the end, so by reading the text in order, there are few new terms that are not explained before. I think the result should be understood, at least broadly, without the need of reading the methods. So, throughout my comments I stress few terms that could be clarified in the results.

Tables don't come in order in the text, again because material and methods are at the end.

Specific comments: The PDF I downloaded had many sentences repeated over and over (an issue in the pdf preparation I assume), so I guess the numbers don't correspond to what the authors uploaded. For this reason I have used the word document and numbered the lines myself, and my comments correspond to that numbering.

To make sure is clear, the introduction starts in line 56, Results in Line 110, Discussion in Line 301 and Material & Methods in Line 420.

Line 133. The figure 2 shows a potential pilus, I think it should be mentioned/described in the results as well.

Line 140. Since Materials and methods are at the end of the text in this journal, "pristine" and "mooring" needs to be explained here somehow or in the introduction.

Line 146. In general I am not sure of the use of "symbiont MAGs", there is no prove that they are real symbionts at this point, maybe they are transiently associated microbiota. This analysis can not prove the 'role' of these bacteria with the host.

Line 148. Patesci_98 seems to be missing in the Table S4. Did I miss something?

Line 155. The amplicon analysis is rather underused in the text. It is only mentioned in the main text

to highlight that the selected MAGs were within the 15 most abundant microbial phyla, which can be seen in a confusing supplementary Figure (see below). There is not information to evaluate this data. Provide a supplementary table with the samples used in amplicon analysis showing total number of sequences, unique ASVs, filtering of ASVs if applies, rarefaction etc, so the reader can trust this plot/result. Maybe you could also present results from seawater and sediments, not only the percentages in the sponge. Something like the Patescibacteria classes (Fig. S3), but for all of them. This would help to see if the 15 most dominant phyla are also dominant in teh environment or not.

Figure S2. Following the previous comment, this is not an easy figure to understand. Does the overlap of some proteobacterial classes mean anything? I did not get the explanation of the asterisk. Wouldn't a table with the values of the relative abundances of groups, and their classification level be easier to understand? Or a basic stacked barplot, also adding environmental samples, if you decide to do so.

Fig 169. Explain here briefly what are the "controls". Also, I see in the Figure S3 that controls must be sediment and seawater, however I don't know where sediments come from. As long as I understand, there are 7 sponge samples (from 2 different locations) and 5 seawater samples for controls (lines 497-502), but I can't see any sediment sample described anywhere. Also, does this heatmap represent the AVERAGE relative abundance of all sample types then? Please clarify all this in text and figures.

Figure 6. Out of curiosity, what is the brownish shape in the background?

Line 201. The "Predicted lifestyle of the major players" section is a very detailed description of functions within the selected MAGs. Assuming that the analysis was done correctly, I have no further comments on this descriptive part (also for the supplementary text about it). I think the authors have selected a good number of important pathways to describe the most important findings, and it is clearly explained.

Line 427. Remove parenthesis et the end of line.

Line 440 vs. Line 497. DNA was extracted twice from the same samples (Table S1). One time with DNeasy Power Soil Kit (Qiagen) for both sponges and water (I am assuming this), and another with QiagenAllPrep DNA/RNA Mini Kit. Is there a reason for this? Couldn't the same DNA extraction be used in both amplicon and metagenomic sequencing?

As I said, the water filters are not differentiated in line 440, were they extracted from the other half of a PVDF membrane filter? Any adaptation of the Power Soil Kit for the filters?

Line 422. Change to eastern Canadian.

Figure S1. I am not familiar with the LEfSE program, but why the LDA scores are negative for vazella in A) but positive for the vazella subsets in B). Clarify whether this needs to be understood as absolute values or implies something else. And what does " An LDA score of 2 was selected as cut-off." mean? Cutoff for what? Is this log₁₀ anyway?

Line 537. How did you convert the Enzyme Commission numbers to K numbers?

Line 538. This link (<https://www.genome.jp/kegg/>) does not bring the KEGG mapper.

Figure 5. Do you mean > 50 % for the solid lines? Line 548. The figshare data seems not available yet.

The work from Kristina Bayer and coauthors entitled "Microbial strategies for survival in the glass sponge *Vazella pourtalesii*" describes the microbial community of the glass sponge *vazella pourtalesii* using microscopic description, amplicon analysis and metagenome-assembled genomes (MAGs) from 7 samples of sponges. The metabolic capacities are described based in K terms and other resources, which is also linked to the microscopic results and amplicon relative abundances.

I find this work to be quite interesting and well executed. Information about glass sponges is more scarce than demosponges, therefore this is a valuable resource for the sponge community. The authors have done a great effort analysing the functional repertory of dominant sponge symbionts, providing a summary in the main text and a detailed description in supplementary material. The manuscript is well-written and Introduction and Discussion are very nicely presented.

Overall, I think this manuscript would be an excellent fit for mSystems, and with some revisions and pending some clarifications from the authors I would recommend acceptance.

My main concern is the specific reason for the selected MAGs. The selection of 13 MAGs is the key point for this study, since everything else is based on those selected ones (Line 141). MAGs are chosen based somehow in their enrichment in sponge samples versus water (LDA scores), in their completeness, and redundancy values, however, I don't see a clear justification for selecting those ones and not others.

For instance, why was bin_27 (the forth one with highest LDA score, 87.9 completeness, and being alphaproteobacteria) not selected but it was bin_136s with 61.2 completeness and lower LDA score? Is there a cutoff of some kind? Is it based on the taxonomy of the MAGs? Was this a personal selection? I wonder if the selection of other MAGs would bring a completely different story.

As a general comment, I want to point out that in this journal Methods are presented at the end, so by reading the text in order, there are few new terms that are not explained before. I think the result should be understood, at least broadly, without the need of reading the methods. So, throughout my comments I stress few terms that could be clarified in the results.

Tables don't come in order in the text, again because material and methods are at the end.

Specific comments: The PDF I downloaded had many sentences repeated over and over (an issue in the pdf preparation I assume), so I guess the numbers don't correspond to what the authors uploaded. For this reason I have used the word document and numbered the lines myself, and my comments correspond to that numbering. To make sure is clear, the introduction starts in line 56, Results in Line 110, Discussion in Line 301 and Material & Methods in Line 420.

Line 133. The figure 2 shows a potential pilus, I think it should be mentioned/described in the results as well.

Line 140. Since Materials and methods are at the end of the text in this journal, "pristine" and "mooring" needs to be explained here somehow or in the introduction.

Line 146. In general I am not sure of the use of "symbiont MAGs", there is no prove that they are real symbionts at this point, maybe they are transiently associated microbiota. This analysis can not prove the 'role' of these bacteria with the host.

Line 148. Patesci_98 seems to be missing in the Table S4. Did I miss something?

Line 155. The amplicon analysis is rather underused in the text. It is only mentioned in the main text to highlight that the selected MAGs were within the 15 most abundant microbial phyla, which can be seen in a confusing supplementary Figure (see below). There is not information to evaluate this data. Provide a supplementary table with the samples used in amplicon analysis showing total number of sequences, unique ASVs, filtering of ASVs if applies, rarefaction etc, so the reader can trust this plot/result. Maybe you could also present results from seawater and sediments, not only the percentages in the sponge. Something like the Patescibacteria classes (Fig. S3), but for all of them. This would help to see if the 15 most dominant phyla are also dominant in teh environment or not.

Figure S2. Following the previous comment, this is not an easy figure to understand. Does the overlap of some proteobacterial classes mean anything? I did not get the explanation of the asterisk. Wouldn't a table with the values of the relative abundances of groups, and their classification level be easier to understand? Or a basic stacked barplot, also adding environmental samples, if you decide to do so.

Fig 169. Explain here briefly what are the "controls". Also, I see in the Figure S3 that controls must be sediment and seawater, however I don't know where sediments come from. As long as I understand, there are 7 sponge samples (from 2 different locations) and 5 seawater samples for controls (lines 497-502), but I can't see any sediment sample described anywhere. Also, does this heatmap represent the AVERAGE relative abundance of all sample types then? Please clarify all this in text and figures.

Figure 6. Out of curiosity, what is the brownish shape in the background?

Line 201. The "Predicted lifestyle of the major players" section is a very detailed description of functions within the selected MAGs. Assuming that the analysis was done correctly, I have no further comments on this descriptive part (also for the supplementary text about it). I think the authors have selected a good number of important pathways to describe the most important findings, and it is clearly explained.

Line 427. Remove parenthesis et the end of line.

Line 440 vs. Line 497. DNA was extracted twice from the same samples (Table S1). One time with DNeasy Power Soil Kit (Qiagen) for both sponges and water (I am assuming this), and another with QiagenAllPrep DNA/RNA Mini Kit. Is there a reason for this? Couldn't the same DNA extraction be used in both amplicon and metagenomic sequencing?

As I said, the water filters are not differentiated in line 440, were they extracted from the other half of a PVDF membrane filter? Any adaptation of the Power Soil Kit for the filters?

Line 422. Change to eastern Canadian.

Figure S1. I am not familiar with the LEfSE program, but why the LDA scores are negative for *Vazella* in A) but positive for the *Vazella* subsets in B). Clarify whether this needs to be understood as absolute values or implies something else. And what does "An LDA score of 2 was selected as cut-off." mean? Cutoff for what? Is this log10 anyway?

Line 537. How did you convert the Enzyme Commission numbers to K numbers?

Line 538. This link (<https://www.genome.jp/kegg/>) does not bring the KEGG mapper.

Figure 5. Do you mean > 50 % for the solid lines?

Line 548. The figshare data seems not available yet.

We would like to thank you and the two reviewers for the valuable comments and suggestions that aided us in improving our manuscript for publication in mSystems. Please find our replies to the individual reviewer comments attached below. The indicated line numbers refer to the marked-up manuscript.

Reviewer comments:

Reviewer #1 (Comments for the Author):

The manuscript 'Microbial strategies for survival in the glass sponge *Vazella pourtalesii*' by Kristina Bayer et al. investigated the composition of the microbiota associated with the glass sponge *Vazella pourtalesii* and the functional strategies of the main symbionts using combined strategy of microscopic approaches with metagenome-guided microbial genome reconstruction and amplicon community profiling. Based on metagenome-assembled genomes analyses, SAR324 bacteria, Crenarchaeota, Patescibacteria and Nanoarchaeota were identified as abundant members of the *V. pourtalesii* microbiome. 13 representative *V. pourtalesii*-enriched MAGs from 137 were selected for detailed metabolic features analyses. Particularly the metabolic interactions between the four microbial taxa were highlighted. It was suggested that Crenarchaeota and SAR324 produce and partly secrete all required amino acids and vitamins. Nanoarchaeota and Patescibacteria, tap into the microbial community for resources, e.g., lipids and DNA, likely using pili-like structures. The results expand our understanding of the interaction between uncultured sponge microbes in a sponge holobiont. It is recommended for publication after minor revision:

1. What are the roles of Nanoarchaeota and Patescibacteria? Only takers? It will strengthen the interaction between the microbes if some functional features of Nanoarchaeota and Patescibacteria were revealed.

The functional features of the “takers” are described in the results and summarized in Fig. 6. We have now included the possible role of Nanoarchaeota in carbon fixation in the discussion as well (l. 388-389).

2. The references format needs to be unified: The first letter of the title words should be lowercase except the first word. References 14, 30, 32,33. 44, 51,53,56, 58,61,64,66,,68,101,102..... are wrong. The format of book references are suggested to be adjusted according to the journal requirement, for example 1,9,12, 31.....

The reference format was fixed.

Reviewer #2 Comments:

The work from Kristina Bayer and coauthors entitled "Microbial strategies for survival in the glass sponge *Vazella pourtalesii*" describes the microbial community of the glass sponge *vazella*

pourtalessi using microscopic description, amplicon analysis and metagenome-assembled genomes (MAGs) from 7 samples of sponges. The metabolic capacities are described based in K terms and other resources, which is also linked to the microscopic results and amplicon relative abundances.

I find this work to be quite interesting and well executed. Information about glass sponges is more scarce than demosponges, therefore this is a valuable resource for the sponge community. The authors have done a great effort analysing the functional repertory of dominant sponge symbionts, providing a summary in the main text and a detailed description in supplementary material. The manuscript is well-written and Introduction and Discussion are very nicely presented.

Overall, I think this manuscript would be an excellent fit for mSystems, and with some revisions and pending some clarifications from the authors I would recommend acceptance. My main concern is the specific reason for the selected MAGs. The selection of 13 MAGs is the key point for this study, since everything else is based on those selected ones (Line 141). MAGs are chosen based somehow in their enrichment in sponge samples versus water (LDA scores), in their completeness, and redundancy values, however, I don't see a clear justification for selecting those ones and not others.

For instance, why was bin_27 (the forth one with highest LDA score, 87.9 completeness, and being alphaproteobacteria) not selected but it was bin_136s with 61.2 completeness and lower LDA score? Is there a cutoff of some kind? Is it based on the taxonomy of the MAGs? Was this a personal selection? I wonder if the selection of other MAGs would bring a completely different story.

Yes, indeed, we first identified the dominant phyla based on the amplicon data (detailed in Busch et al., BioRxiv DOI 10.1101/2020.05.19.102806) and the metagenomic data presented here. Then all sponge-enriched MAGs for these phyla were selected for detailed analysis.

Rather than a sweeping, superficial analysis of all MAGs, we opted for an in-depth analysis of selected, dominant MAGs representative of the microbial consortium. It can not be excluded that some information has been missed.

As a general comment, I want to point out that in this journal Methods are presented at the end, so by reading the text in order, there are few new terms that are not explained before. I think the result should be understood, at least broadly, without the need of reading the methods. So, throughout my comments I stress few terms that could be clarified in the results. Tables don't come in order in the text, again because material and methods are at the end.

Thanks for pointing this out. We added short explanations in the 'MAG selection' part of the results section (l. 131-132), moved acronym descriptions to the respective first occurrence in the text, and we corrected the numbering of the supplementary tables.

Specific comments: The PDF I downloaded had many sentences repeated over and over (an issue

in the pdf preparation I assume), so I guess the numbers don't correspond to what the authors uploaded. For this reason I have used the word document and numbered the lines myself, and my comments correspond to that numbering.

To make sure is clear, the introduction starts in line 56, Results in Line 110, Discussion in Line 301 and Material & Methods in Line 420.

Line 133. The figure 2 shows a potential pilus, I think it should be mentioned/described in the results as well.

The potential pili are mentioned in the results now (l. 128).

Line 140. Since Materials and methods are at the end of the text in this journal, "pristine" and "mooring" needs to be explained here somehow or in the introduction.

The different sampling locations are mentioned in the "MAG selection" part of the results now (l. 131-132).

Line 146. In general I am not sure of the use of "symbiont MAGs", there is no prove that they are real symbionts at this point, maybe they are transiently associated microbiota. This analysis can not prove the 'role' of these bacteria with the host.

Agreed! We have specified our terminologies in that we use the term "symbiont" when it applies to the well documented shallow water sponge symbioses literature, but have deleted the term "symbiont MAGs" from this manuscript. However, in the discussion we hypothesize that the analyzed microbes of *Vazella* may indeed be symbionts and provide a rationale for this hypothesis (l. 367-369).

Line 148. *Patesci_98* seems to be missing in the Table S4. Did I miss something?

Patesci_98 was added to Table S4.

Line 155. The amplicon analysis is rather underused in the text. It is only mentioned in the main text to highlight that the selected MAGs were within the 15 most abundant microbial phyla, which can be seen in a confusing supplementary Figure (see below). There is not information to evaluate this data. Provide a supplementary table with the samples used in amplicon analysis showing total number of sequences, unique ASVs, filtering of ASVs if applies, rarefaction etc, so the reader can trust this plot/result. Maybe you could also present results from seawater and sediments, not only the percentages in the sponge. Something like the *Patescibacteria* classes (Fig. S3), but for all of them. This would help to see if the 15 most dominant phyla are also dominant in teh environment or not.

We have reduced the amplicon analysis purposely to a minimum, as we have submitted a separate manuscript with a focus on 16S rDNA data of *Vazella pourtalesii*, along with seawater and sediment references (Busch et al., BioRxiv DOI 10.1101/2020.05.19.102806), in which we

characterize the microbial community composition and assess its variability in response to anthropogenic impacts. The respective manuscript has been made available on BioRxiv and is referenced as such throughout the present manuscript.

Figure S2. Following the previous comment, this is not an easy figure to understand. Does the overlap of some proteobacterial classes mean anything? I did not get the explanation of the asterisk. Wouldn't a table with the values of the relative abundances of groups, and their classification level be easier to understand? Or a basic stacked barplot, also adding environmental samples, if you decide to do so.

We have modified Figure S2 by plotting the circles separately and removing the asterisk.

Fig 169. Explain here briefly what are the "controls". Also, I see in the Figure S3 that controls must be sediment and seawater, however I don't know where sediments come from. As long as I understand, there are 7 sponge samples (from 2 different locations) and 5 seawater samples for controls (lines 497-502), but I can't see any sediment sample described anywhere. Also, does this heatmap represent the AVERAGE relative abundance of all sample types then? Please clarify all this in text and figures.

The details of sampling sponges, seawater and sediment controls are provide in reference 20 (Busch et al., BioRxiv DOI 10.1101/2020.05.19.102806) (lines 420-421). Here, we provide a brief summary of the procedures.

Figure 6. Out of curiosity, what is the brownish shape in the background?

These are the outlines of a *Vazella pourtalesii* sponge, but we understand that this was not clear enough. The image has been redrawn and the legend has been adapted for more clarity, the contents of the figure remain unchanged.

Line 201. The "Predicted lifestyle of the major players" section is a very detailed description of functions within the selected MAGs. Assuming that the analysis was done correctly, I have no further comments on this descriptive part (also for the supplementary text about it). I think the authors have selected a good number of important pathways to describe the most important findings, and it is clearly explained.

We are very happy to hear this – thank you for this comment!

Line 427. Remove parenthesis et the end of line.

We corrected this sentence (l. 420-423).

Line 440 vs. Line 497. DNA was extracted twice from the same samples (Table S1). One time with DNeasy Power Soil Kit (Qiagen) for both sponges and water (I am assuming this), and another with QiagenAllPrep DNA/RNA Mini Kit. Is there a reason for this? Couldn't the same DNA extraction be used in both amplicon and metagenomic sequencing?

As I said, the water filters are not differentiated in line 440, were they extracted from the other half of a PVDF membrane filter? Any adaptation of the Power Soil Kit for the filters?

We used a standardized protocol for amplicon sequencing with the DNeasy kit. We had further extracted nuclei acids (DNA/RNA) for meta-omics using the QiagenAllPrep DNA/RNA co-extraction kit. The metatranscriptomic part was dropped from this project due to low RNA yields. We suggest that this explanation does not need to be elaborated in the manuscript.

As for the water filters, two halves of the same filter were used for either amplicon or metagenomic DNA extraction, this is stated more clearly now (l. 497-499). No further adaptation for the Power Soil Kit for the filters were done, which is not worth mentioning in our opinion.

Line 422. Change to eastern Canadian.

We added a space between “Canada” and “in” instead (l. 417)

Figure S1. I am not familiar with the LEfSE program, but why the LDA scores are negative for vazella in A) but positive for the vazella subsets in B). Clarify whether this needs to be understood as absolute values or implies something else. And what does " An LDA score of 2 was selected as cut-off." mean? Cutoff for what? Is this log10 anyway?

Figure S1A and the corresponding legend were adapted.

Line 537. How did you convert the Enzyme Commission numbers to K numbers?

The KEGG Orthology (KO) reference hierarchy was used (e.g., https://www.genome.jp/kegg-bin/get_htext#C1) to convert EC to K numbers with an in-house R script. A statement was added to the methods section (l. 531-536).

Line 538. This link (<https://www.genome.jp/kegg/>) does not bring the KEGG mapper.

The link was corrected (l. 536).

Figure 5. Do you mean > 50 % for the solid lines? Line 548. The figshare data seems not available yet.

The figure legend was corrected and the figshare data is publicly available now.

July 23, 2020

Dr. Beate M Slaby
GEOMAR Helmholtz Centre for Ocean Research Kiel
Marine Symbioses
Düsternbrooker Weg 20
Kiel 24105
Germany

Re: mSystems00473-20R1 (Microbial strategies for survival in the glass sponge *Vazella pourtalesii*)

Dear Dr. Beate M Slaby:

Thank you very much for your revision and attention to the reviewer's comments. Your manuscript has been editorially accepted, and I am forwarding it to the ASM Journals Department for publication. For your reference, ASM Journals' address is given below. Before it can be scheduled for publication, your manuscript will be checked by the mSystems senior production editor, Ellie Ghatineh, to make sure that all elements meet the technical requirements for publication. She will contact you if anything needs to be revised before copyediting and production can begin. Otherwise, you will be notified when your proofs are ready to be viewed.

Sincerely,

Seth Bordenstein
Editor, mSystems

Journals Department
Phone: 1-202-942-9338